# Gut health predictive indices linking gut microbiota dysbiosis with healthy state, mild gut discomfort, and inflammatory bowel disease phenotypes using gut microbiome profiling

Joann Phan,[1] Suneer Jain,[1] Jurgen F. Nijkamp,[2] Rajkumar Sasidharan,[3] Ashish Agarwal,[3] Julia K. Bird,[4] Anneleen Spooren,[5] Jonas Wittwer Schegg,[5] Emiel Ver Loren van Themaat,[2] Tim N. Mak[5]

**ABSTRACT**  Despite the complexity of the gut microbiome, several scores that use taxonomic characteristics exist that attempt to identify a healthy gut or gastrointestinal disease. Two systems in use are the metagenomic aerotolerant predominance index (MAPI) and keystone scores. The aim of this analysis was to compare different gut microbiome scores, specifically MAPI and a keystone species score, on two cross-sectional data sets and to investigate correlations of these scores with self-reported gut discomfort and gastrointestinal disease. The first data set is a commercial data set (Sun Genomics data set) with whole-genome shotgun sequencing samples from 5,372 customers. The second data set is curated from publicly available data (public data set) with 2,415 samples from participants in human studies with gut-related taxonomic profiles. MAPI scores and keystone species scores were calculated using standard methodology. The MAPI score was significantly lower in men for the public data set. There was a graded response for both the MAPI and keystone scores between healthy subjects, subjects with mild gastrointestinal discomfort, and patients with gastrointestinal disease: the MAPI score was higher, and the keystone score was lower in subjects with gastrointestinal discomfort or with inflammatory bowel disease patients. The keystone and MAPI scores have the potential to help identify factors associated with gut microbial dysbiosis and gastrointestinal discomfort or disease. Furthermore, given the functional link of the MAPI score to oxidative stress in the microbiome, the scores can help to identify conditions where oxidative stress is one of the hallmarks of dysbiosis.

**IMPORTANCE**  Gut bacteria play a role in both mild gastrointestinal discomfort, which includes bloating and constipation, and inflammatory bowel disease. There are many different types of bacteria in the gut, and gut microbiome composition differs greatly between different people. Therefore, it is difficult to predict who has a gut microbiome associated with a healthy gut and who might develop disease or experience gut discomfort. Several scoring systems have been developed to categorize gut health states. This analysis compared two different scoring systems using data from two different sources to see how well they could identify people with gastrointestinal disease, gastrointestinal complaints, or a healthy gut. The scoring systems showed similar trends according to gut health status: groups of people with gut bacteria imbalance or gut disease had a different score than groups of people with healthy gut bacteria.

**KEYWORDS**  gut microbiome, inflammatory bowel disease, gut inflammation, keystone score, metagenomic aerotolerant predominance index

The complex microbial communities in the human digestive tract known as the gut microbiota are intricately involved in human health and disease (1). Considerable

**Peer Reviewer** Asaduzzaman Asad, International Centre for Diarrhoeal Disease Research, Dhaka, Bangladesh

Address correspondence to Tim N. Mak, tim.mak@trilliome.com, Emiel Ver Loren van Themaat, emiel.ver-loren-van-themaat@dsm-firmenich.com, or Jonas Wittwer Schegg, jonas.wittwer@dsm-firmenich.com.

The authors declare no conflict of interest

research has focused on the direct effects of the gut microbiome on gastrointestinal health, with emphasis on disease states and their association with microbial dysbiosis. A result of that research is the concept of gut microbiome homeostasis associated with health: disease or other gastrointestinal symptoms are thought to arise from an imbalance of the gut microbiome or an increased abundance of pathogens (2). Dysbiosis is implicated in mild gastrointestinal conditions, such as irritable bowel syndrome (IBS; [3]) and inflammatory bowel disease (IBD; [4]). There is nevertheless considerable uncertainty in the role of dysbiosis in human health: microbial imbalance of the gut has not unequivocally been shown to be a cause of disease and not a result of poor health. Moreover, there is a need to move beyond the demonstration of dysbiosis and toward the development of predictive tests or treatments for microbiome-related disease (5).

Diagnostic tests based on stool microbiome samples could potentially assist in the detection of microbial dysbiosis or to identify microbial features associated with gastrointestinal discomfort or gastrointestinal disease. Many different indices are currently under development that could use gut microbiome profiles to gauge the health and functionality of the microbiome and the potential for the development of disease (2). Indices based on alpha diversity are already associated with gastrointestinal disease status, such as the commonly used Shannon index (6), which considers the number of organisms within the ecosystem and the uniformity of population size of each species (7). There are several other indices of note that are currently under development that aim to bridge gut composition to functionality and health or disease: the metagenomic aerotolerant predominance index (MAPI) and keystone scores are two examples of these.

The metagenomic aerotolerant predominance index was originally created by linking the gut redox potential to acute malnutrition (8). It is defined as the natural logarithm of the ratio of the relative abundance of aerotolerant to strict anaerobic species (8). A MAPI score greater than 0 indicates predominance of aerobes, while a negative score is associated with a higher relative abundance of anaerobes. Various publications examining gut dysbiosis have shown overgrowth of aerotolerant or facultative aerobic species as a sign of gut microbiome dysbiosis and gastrointestinal disorders (9). For example, gut microbiome dysbiosis was observed following antibiotic treatment characterized by aerobic *Salmonella*-induced gastroenteritis. Antibiotics induced a depletion of butyrate-producing Clostridia, which is associated with negative effects on gut homeostasis (10–12). Another example is the overgrowth of commensal *Escherichia coli* in conditions of intestinal inflammation following host-driven inflammatory responses that generate reactive oxygen and nitrogen species, which conferred a growth advantage to *E. coli* (13). However, the complexity of the gut microbiome means that it is difficult to translate measures of diversity or the presence of certain microbes to a robust score.

Similarly, the keystone species approach uses a limited number of eight common strictly anaerobic gut commensals to define a healthy gut microbiome (14, 15). Under dysbiosis of the gut microbiota, certain functions exerted by the microbiome are lost, such as adequate gut barrier function, resulting in adverse effects on host health (16). Keystone species are postulated to be species that carry unique functions that are essential for the balance of the gut microbiota (15). These species exert a beneficial effect on gut balance and functionality; they are acting as "ecosystem engineers," and when lost, they disproportionally lead to poorer gut health (15). Identifying keystone species functions across health and gastrointestinal disease could help to identify targeted interventions to selectively support their growth, functionality, and abundance in the gut microbiome ecosystem.

The ability of pathogens to cause dysbiosis might be linked at least, in part, to their ability to deal better with oxidative stress than commensal anaerobes. Indeed, 8 among the 12 WHO-listed antibiotic-resistant priority pathogens are facultative aerobes (17). The ability of pathogens to trigger metabolic pathways that allow them to handle oxidative

stress might have given pathogenic species an evolutionary advantage, allowing them to outgrow commensal anaerobes in circumstances of competition for limited resources in the gut or to "hijack" the host inflammatory response, including oxidative stress, to their advantage. Conversely, the ability of commensal anaerobes to deal with oxidative stress could contribute to health and gut homeostasis (18). A recent example includes the correlation of the presence of gut-microbiome-resident antioxidant systems to health and longevity in centenarians in the Jiaoling area in China (19). Another example is the association observed between response to beta-fructan fibers in ulcerative colitis patients, where patients with an absence of gut microbiome production of riboflavin, a key redox-active and antioxidant vitamin in the gut, exhibited a proinflammatory response to beta-fructan fibers (20, 21).

These diverse examples illustrate the potential importance of functional oxidative stress response pathways in commensal anaerobes and the balance between aerotolerant and strict anaerobic species for maintaining microbiome homeostasis. While the focus has been mostly on metagenomic and metabolomic output in gut microbiome research (22), the understanding of the role of the gut environment and oxidation levels in gut homeostasis and dysbiosis remains limited. Moreover, redox balance is also linked to bioenergetic flows and cross-feeding in the microbiome (23), and therefore, any changes in redox balance or oxidative stress could trigger systemic effects on the gut microbiome ecosystem via changing the cross-feeding flows (18). It is well known from research into human homeostasis that the adaptive response to oxidative stress is one of the characteristics underlying homeostasis (24), but the understanding of the oxidative stress response in the gut microbiome and its role in gut microbiome homeostasis remains more limited (25). Elucidating the correlation of imbalance between facultative aerobe and strict anaerobic species, as described by the MAPI index on the one hand (8) and gastrointestinal discomfort and disease on the other, could be of special interest.

Therefore, the MAPI index and keystone species can be seen as two distinct markers of gut dysbiosis. Our research aimed to further map how these markers could link gut dysbiosis to gut discomfort and gastrointestinal disease. First, we investigated the use of several gut microbiome indices, derived from metagenomic analyses, to differentiate between healthy, mild gut discomfort and diseased populations, within major demographic determinants using publicly available and commercial data sets. Second, we compared the performance of the indices with each other.

The aims of further characterizing gut health indices in large-scale population cohorts, as described here, are (i) establishing characteristics that can be correlated to mild gut discomfort or gastrointestinal disease, (ii) identifying characteristics that could be modulated by interventions to restore health and gut homeostasis, and (iii) deepening our understanding of fundamental modes of action and mechanistic changes that underlie gut dysbiosis and gastrointestinal discomfort and disease.

## RESULTS

### Sun Genomics customer database analysis

In total, 5,372 profiles were used for the analysis, and demographic information is presented in Table 1. More participants were female than male, and gender was unknown for one-fifth of participants. Most participants were adults aged between 21 and 60 years, with a small proportion of children and adolescents, or seniors. Participants were from 40 countries overall; however, most participants were located in the US.

MAPI score was analyzed according to gender and age categories in the Sun Genomics data set (Fig. 1). The average MAPI score was not significantly different in male compared to female participants (Wilcoxon signed-rank test, $P > 0.05$; Fig. 1A). MAPI score increased with age (Fig. 1B and C), and there were significant differences found between different age categories (Fig. 1D). Namely, the MAPI scores of age category 1–10 were significantly lower than 61–80 years, 11–20 years were significantly lower than 51–80 years, and age category 21–60 years was significantly lower than the age category

**TABLE 1** Sun Genomics customer and public database demographics

| Variable | N | % | N | % |
|---|---|---|---|---|
| Data set | Sun Genomics | | Public data set | |
| Total number of participants | 5,372 | | 2,067 | |
| Gender—male | 1,979 | 36.8% | 1,075 | 52% |
| Gender—female | 2,673 | 49.7% | 441 | 21.3% |
| Gender—not stated (%) | 1,112 | 20.6% | 551 | 13.5% |
| Age 0–10 yr | 602 | 11.2% | 203 | 9.8% |
| Age 11–20 yr | 331 | 6.2% | 425 | 20.6% |
| Age 21–30 yr | 640 | 11.9% | 287 | 13.9% |
| Age 31–40 yr | 1,213 | 22.6% | 558 | 27% |
| Age 41–50 yr | 1,045 | 19.5% | 10 | 0.5% |
| Age 51–60 yr | 846 | 15.7% | 51 | 2.5% |
| Age 61–70 yr | 476 | 8.9% | 63 | 3% |
| Age 71–80 yr | 189 | 3.5% | 16 | 0.8% |
| Age 81+ yr | 30 | 0.6% | – | – |
| Country—US | 4,909 | 91.4% | – | – |
| Country—UK | 162 | 3.0% | – | – |
| Country—other | 301 | 5.6% | – | – |

61–80 years. The small sample size for the 91–100-year category prevents generalizations from being made about this age group.

The MAPI score was compared between healthy, mild, and diseased phenotypes in Fig. 2. MAPI score was higher in the mild and disease phenotypes compared to the healthy phenotype, indicating that participants with the greatest disease burden had a slightly higher relative abundance of aerotolerant microbes. The Kruskal-Wallis test indicates that at least one phenotype stochastically dominates the other(s). When assessing the alpha and beta diversities of the profiles across phenotypic bins, there was no significant difference in alpha diversity and low separation between bins on a Principal Component Analysis (results not shown).

Disease phenotypes were compared using the keystone species score and individual keystone species abundance (Fig. 3). $Log_{10}$ bin sum of the keystone species score is decreased in the diseased phenotype compared to higher scores in the healthy phenotype, indicating that there is a higher proportion of keystone species in the healthy population and a lower proportion of keystone species in the diseased population (Fig. 3A). The relative proportion of individual keystone species was the highest in the healthy phenotype, except for *Methanobrevibacter smithii* (Fig. 3B).

## Public data set analyses

A total of 2,415 MAPI scores from healthy adults were collated from 8 independent studies in order to obtain an overall healthy MAPI score distribution (Table 2). From these, 1,075 MAPI scores were from male and 441 were from female participants (Table 1). MAPI score distribution in the public data sets is shown in Fig. 4. On average, the MAPI score of healthy individuals was −5.2 with an SD of 1.8. MAPI scores in healthy males were slightly but significantly lower than in females ($P < 0.0001$).

## Comparison of MAPI score in the Sun Genomics and public data set

The MAPI score in healthy subjects and patients with IBD was compared between the Sun Genomics and public data sets (Fig. 5). Both data sets showed the same significant relationship, with a higher MAPI in the IBD patients compared to healthy controls. However, the average MAPI score was different between the data sets, with a broader range found in the public data set.

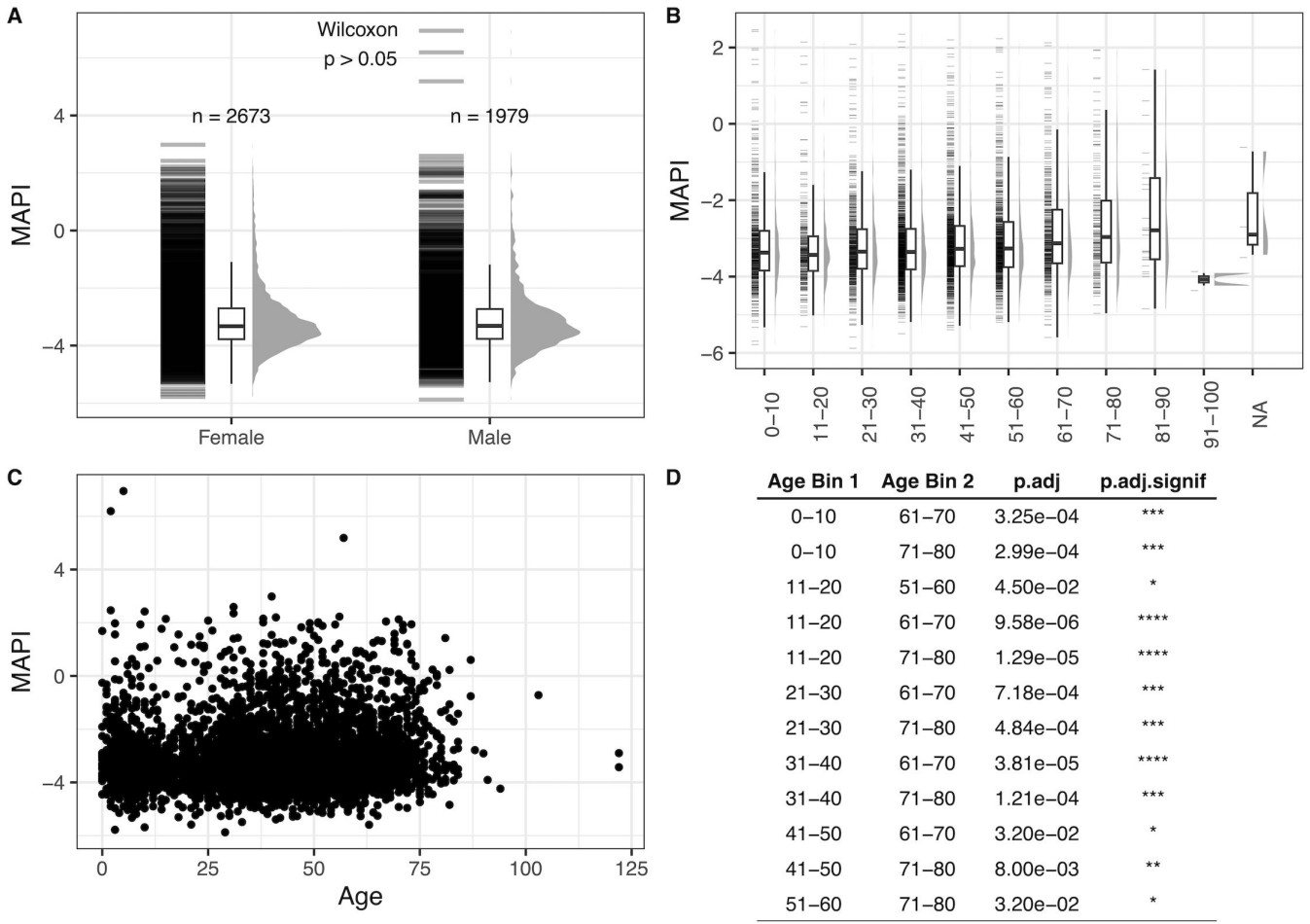

**FIG 1** Age, gender, and MAPI distribution in the Sun Genomics data set. (A) Gender distribution of MAPI score in timepoint one profiles, (B) boxplots of age bins, (C) distribution of MAPI score by age, and (D) MAPI distribution across age bins. Wilcoxon tests were performed to test significance between MAPI scores across age bins. *P*-values were corrected for multiple comparisons with False Discovery Rate (FDR) for the results in D.

## DISCUSSION

The current analysis investigated two different indices of gut microbiome health in two data sets. Changes in the scores according to demographic factors and phenotypic differences in terms of chronic gastrointestinal discomfort or disease status were also explored. Significant differences according to gender were only found in the public data set, with a lower mean MAPI score found in men, although the results here may have been skewed due to unbalanced sample inclusion. The Sun Genomics data set found a significant relationship between MAPI score and age; older participants had a higher MAPI score compared to younger age groups. Differences were found in both the MAPI and keystone scores according to disease-related phenotype, with a graded increase through healthy participants, those with mild gastrointestinal conditions, and patients with gastrointestinal disease. A difference was found between healthy subjects and those with IBD when the MAPI score was compared, which was true across data sets. Thus, we found that higher MAPI scores, representing a higher presence of aerotolerant species and possibly higher levels of oxidative stress, were indicative of gastrointestinal mild discomfort and disease, with disease states being characterized by the highest MAPI scores.

The Sun Genomics data set comprised a large number of participants obtained from the customer database; thus, the microbiome scores obtained from this data set give a good indication of the range of values that can be found under the free-living

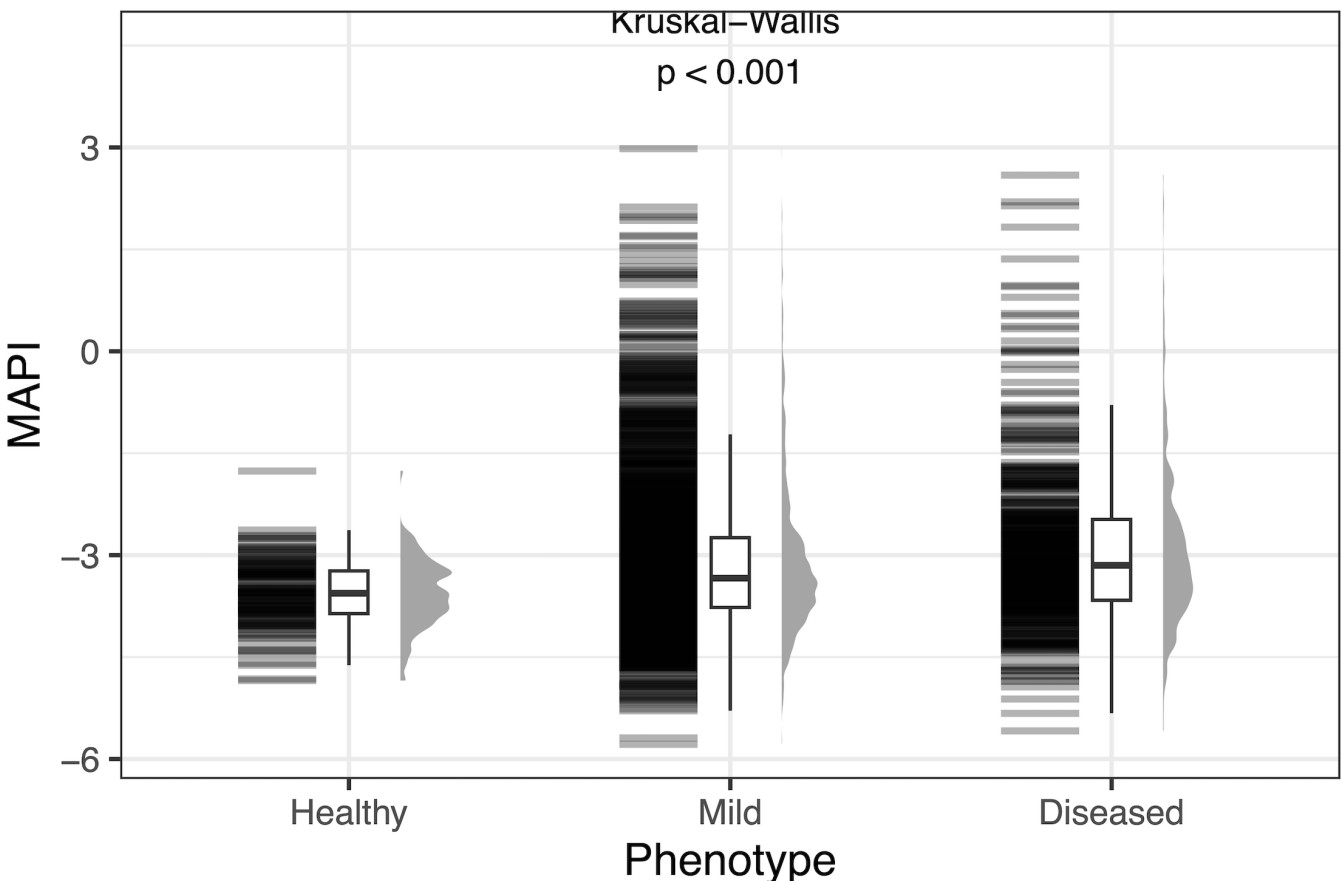

**FIG 2** MAPI score across healthy, mild, and diseased phenotypes in the Sun Genomics data set. The Kruskal-Wallis test was used to calculate the *P*-value.

population in the US, albeit those who have the interest in and ability to purchase a microbiome analysis kit. US-based research has found socio-economic status (SES) to be a strong predictor of the gut microbiome (35), which means that the sample is not representative of the US population or their microbiota, as participants were more likely to have a higher SES. Likewise, the public data set is drawn from a diverse range of clinical trials and cohort studies conducted in various countries, including Germany, Spain, Denmark, France, US, and Canada. The results from this analysis reflect normal scores that can be obtained in surveys and gut microbiome profiles from whole-genome sequencing or amplicon sequencing.

There was no significant difference found in the MAPI score by gender in the Sun Genomics data set, while there was a significant difference found in the public data set. This may reflect bias in the sample. To illustrate, the study described by Turpin and co-authors contained a substantial number of samples from family members (30). The sampling of subjects that share environmental factors may have impacted the correlation between gender and MAPI score (36).

The relationship between MAPI score and age was shown in both data sets with an overall increase in the MAPI score in older subjects (greater than 61 years old). Additionally, the same trend was found by analyzing data from a survey of microbiome samples obtained from 367 Japanese subjects aged 0–104 years, with the highest MAPI score found in the highest age groups (37). An analysis of the genera in samples from the same cohort found a continuous progression in 36 species across the spectrum of aging, with changes in abundance at the highest and lowest age groups (38). Interestingly, this study can be linked to another recent observation of a higher presence of gut-microbiome-resident antioxidant systems in centenarians with a healthy lifespan in

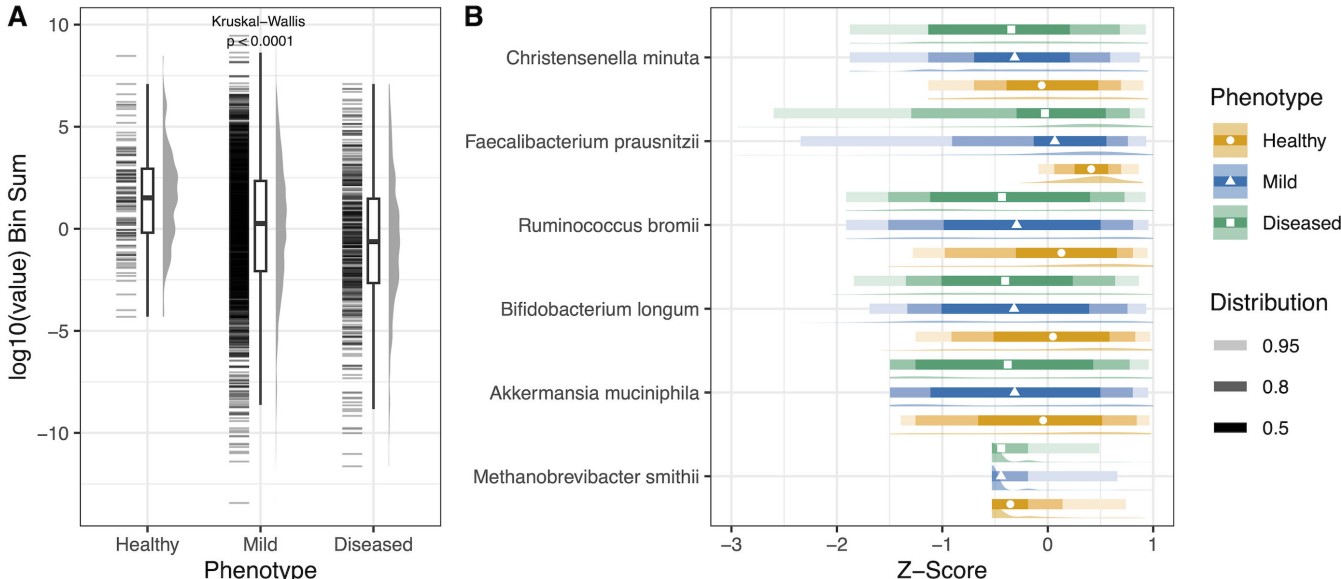

**FIG 3** Keystone species score across phenotypic bins in the Sun Genomics data set. (A) Total mean keystone score by phenotype and (B) individual keystone species relative proportions by phenotype. Relative proportions were log10 transformed, and Z-scores were calculated for each of the six keystone species. The Kruskal-Wallis test was used to calculate the P-value.

the Jiaoling area in China (19). Another possibility is that an increase in medication use for co-morbidities, which is common for chronic conditions in the elderly, induces changes in the microbiome that affect the MAPI score (39). For example, proton pump inhibitors used to treat gastric reflux induce dysbiosis and gastrointestinal conditions, such as small intestine bacterial overgrowth and inflammatory bowel diseases (39). These combined observations provide a basis to further investigate the role of oxidative stress and aerotolerant/strict anaerobe balance in healthy aging. In addition, chronological age may be differentiated from biological age in regard to the microbiome; indeed, microbial imbalance has been identified as a hallmark of aging (40). Metatranscriptomic data have been used to find associations between biological age and the microbiome (41). Future analyses should consider how biological age affects or is affected by the microbiome.

Disease status affected both the MAPI and keystone scores: both showed a gradual change between healthy, mild, and diseased phenotypes, although there was considerable variation, and the differences were small and did not allow individuals to be classified. Figure 2 and 3A show that both scores ranked the healthy cohort lowest, with the mild condition intermediate, and the highest score was found in the diseased cohort. Visually, variation appeared to increase in diseased participants, although this was not tested with an appropriate statistical test. In the original MAPI analysis, the healthy

**TABLE 2** Public data set description

| Study ID | Study description | $N^a$ | Database | Cohort | Reference |
|---|---|---|---|---|---|
| MGYS00005259 | Prebiotic fiber supplementation in 174 healthy young adults (US) | 174 | MGnify | Healthy | (26) |
| MGYS00005601 | 94 Healthy controls from 10 Parkinson's disease trials | 92 | MGnify | Healthy | (27) |
| MGYS00001175 | 26 Healthy young adults given an antibiotic or controls (Canada) | 348 | MGnify | Healthy | (28) |
| MGYS00005628 | 358 Healthy adult controls in a study with colorectal cancer cases (France, Germany, Denmark, and Spain) | 288 | MGnify | Healthy | (29) |
| MGYS00005184 | Large cohort of 1,651 healthy adults, some of whom are related (Canada) | 959 | MGnify | Healthy | (30) |
| Gevers2014_IBD | Healthy controls in four pooled IBD studies | 113 | MicrobiomeHD | Healthy + IBD | (31) |
| Morgan2012_IBD | | | MicrobiomeHD | Healthy + IBD | (32) |
| Papa2012_IBD | | | MicrobiomeHD | Healthy + IBD | (33) |
| Willing2009_IBD | | | MicrobiomeHD | Healthy + IBD | (34) |

$^a$Number of samples.

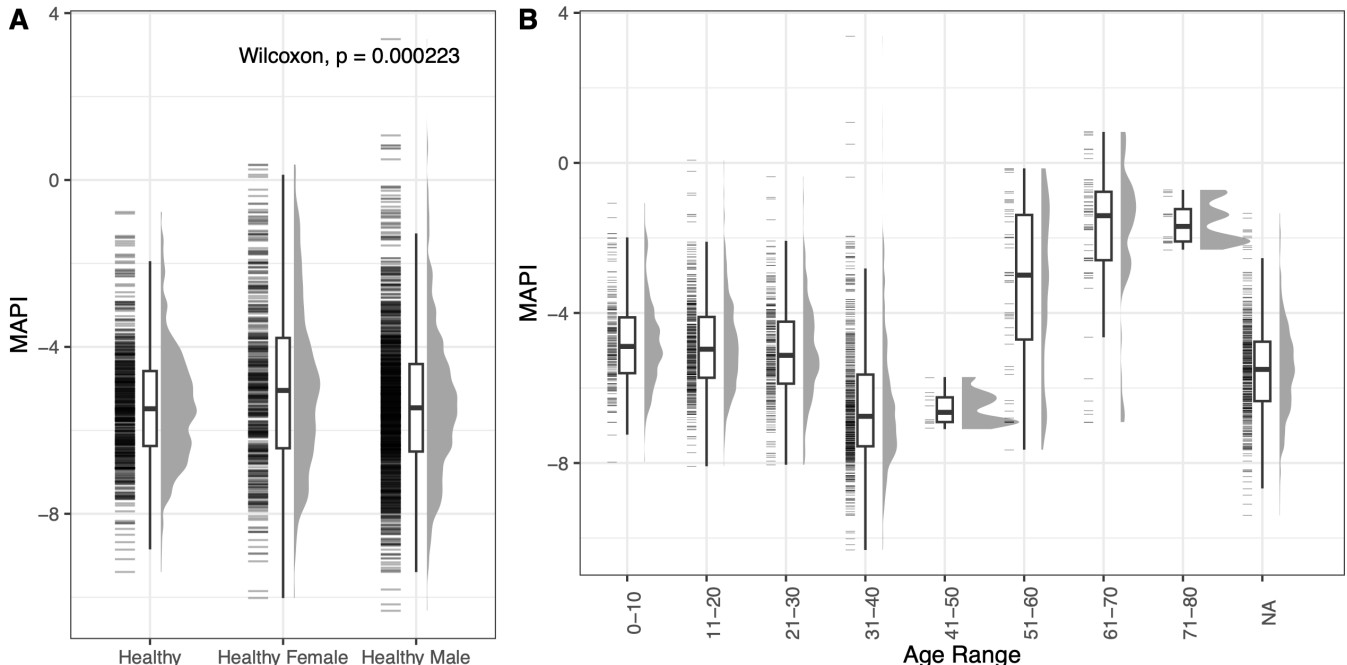

**FIG 4** MAPI score by gender and age category in the public data set. Distribution of MAPI score by (A) overall healthy, healthy males, and healthy females and (B) age categories. The Wilcoxon test was used to calculate the *P*-value.

cohorts of French adults and Senegali children had a lower MAPI score than the patients with dysfunctional dietary status (8). The higher MAPI score could be indicative of oxidative stress and thus a cause or result of the disease condition (8). This investigation further expands the understanding of the correlation of the MAPI index and oxidative stress (9) to gastrointestinal discomfort and disorders.

Despite similar trends seen between the analyses performed on the Sun Genomics and public data sets, differences are seen in the absolute scores found, and the lack of effect by gender seen in the Sun Genomics data set. Methodological differences in the analyses presented here could be responsible for differences in absolute scores. The Sun Genomics data set used whole-genome shotgun sequencing, while the 16S rRNA gene-based and shotgun methods were used for the public data set analysis. These two methods have intrinsic differences in terms of the type and quantity of information collected on the microbiome (42, 43). To our knowledge, there has been no comparison of these two technologies on MAPI score calculation; therefore, it is difficult to specify expected differences. However, identification of bacteria to species level using whole-genome shotgun sequencing will not change the MAPI score calculation, which uses genus information.

While providing a simple way to compare microbiomes, the keystone approach has several drawbacks (44). Using a taxonomic approach does not take into account the effect of microbes that fulfill a redundant functional role in the gut environment yet are not included in the score calculation, which is based on taxonomic classification. Also, it is questionable whether the small number of species selected can account for the complexity found in gut ecosystems, especially when considering inter-individual variation (45). Some species are not included in the keystone approach despite being considered crucial for essential functions in microbiome metabolic competence, such as *Blautia* spp., which is a major electron sink (23). Future work on the keystone score could include this species, given its role within the gut microbial ecosystem. Nevertheless, the observations here linking keystone score with disease state provide a basis for further research in this area.

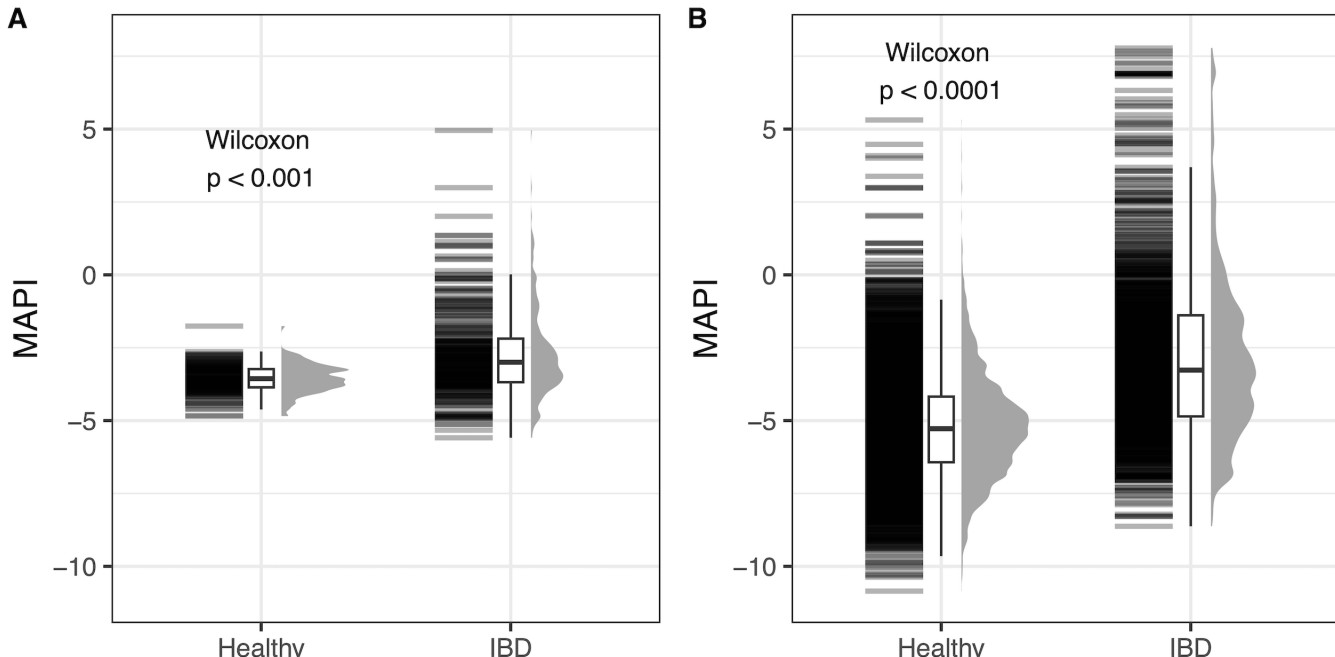

**FIG 5** Comparison of MAPI scores between healthy and IBD subjects in the (A) Sun Genomics and (B) public data sets. MAPI scores for healthy subjects and subjects with IBD in the (A) Sun Genomics and (B) public data sets. The nonparametric Wilcoxon test was used to calculate the *P*-value.

There are several limitations in the data sets that limit the applicability of these results to others. While we wanted to investigate gut microbiome scores within healthy populations, both data sets are drawn from populations with specific limitations. The Sun Genomics data set is drawn from customers that have purchased a service to analyze their microbiome; thus, these participants are likely to be interested in their health and have the means to incur out-of-pocket health-related costs. The demographic and health-related data presented in the Sun Genomics data set also rely completely on self-reported information by participants. A limitation of the self-reported information is the possibility of individual subjectiveness in reporting gastrointestinal health status. Approximately 20% of the participants did not report their gender, indicating that a considerable portion of the demographic data was censored through nonreport for this variable. Despite the Sun Genomics sample being predominantly from the US, the results are unlikely to be representative of the US population because healthy participants who are unable to or not interested in obtaining a microbiome sample were not included. The public data set also suffers from selection bias, including healthy subjects in more highly defined clinical trials who are global but not representative of the entire healthy population.

While this analysis provides some evidence that both the MAPI and keystone scores can partially differentiate between healthy subjects and patients with gastrointestinal symptoms and conditions, the wide variance and overlap in confidence intervals do not allow diagnosis of disease based on the current scores. Both scores rely on taxonomic classification at the genus or species level. Integrated approaches using multi-omics data and separation to entero-signatures during dysbiosis (46) may provide a more robust means to identify microbial dysbiosis that leads to disease and a functional understanding of disease-associated changes in the genome or metabolome (47). The observations on MAPI score and its link to age and gastrointestinal complaints are in line, however, with other studies that investigated the role of oxidative stress and overgrowth of aerotolerant species to microbiome dysbiosis (18).

Our analyses provide a comparison of two gut microbiome scores and how they vary according to demographics and gastrointestinal disease phenotypes. Both gut

microbiome scores showed a differential effect of disease phenotype on the score, which provides a method to distill the complexity of the microbiome into a single score that is relevant to health, although this currently lacks the precision required for clinical practice. Given the general correlation between the MAPI score with redox level and oxidative stress in the intestinal environment, and the correlation of MAPI to gastrointestinal discomfort and disease, the current results suggest that the overgrowth of facultative aerobic/aerotolerant bacteria could be an underlying sign of gut microbiome dysbiosis and is weakly correlated to reported gastrointestinal complaints. Further work is needed to develop diagnostic tools for clinicians to use in diagnosing or treating gastrointestinal disease; however, it is possible that a gut microbiome-based score could be used in the future.

## MATERIALS AND METHODS

### Sun Genomics data set

#### *Participants and sample collection*

The basis of the Sun Genomics data set was the customer database containing cross-sectional information about age, gender, geographic location, a fecal microbiome profile, and a completed health and diet survey.

Fecal samples were taken using at-home collection components for stool with a dry and liquid-filled collection tube and a collection swab that subjects had purchased as part of the Floré Gut Health Test, as described previously (48). Briefly, participants were instructed to use the swab to collect a pea-sized sample of stool immediately after a bowel movement. Participants were asked to ensure that the stool sample was not contaminated with urine. After collection, participants shipped their samples at ambient temperature using the 2-day standard postal service delivery. Upon arrival at Floré laboratories, samples were stored at 4°C and processed within 3 days. When registering their kit, participants provided informed consent for use of data in accordance with IRB no. SG-04142018-001 and were given the opportunity to fill out a health and diet survey.

#### *Microbiome analysis*

Total DNA was extracted and purified using a proprietary method (patents 10428370 and 10837046). To prepare for whole-genome shotgun metagenomics, DNA libraries were prepared with NEBNext reagents and MagJet magnetic beads. Briefly, DNA was sheared, ends were repaired, adapters were ligated, and library concentrations were quantified by qPCR. After library normalization, libraries were pooled and loaded onto an Illumina NextSeq 550 using $150 \times 150$ bp paired-end reads. After sequencing, reads were quality filtered and processed to remove human reads. Taxonomy of the quality-filtered reads was classified using the Gutbuster platform with a hand-curated database of approximately 23,000 microbial species.

#### *Selection of profiles and statistical analysis*

Microbiome profiles with associated health and diet surveys from the Sun Genomics customer database were sorted into healthy, mild, and diseased populations according to self-report by participants in the Sun Genomics health and diet survey. Healthy subjects reported no health or gut issues. Mild conditions included bloating, constipation, gassiness, and IBS. Diseased conditions included Celiac disease, Crohn's disease, gastroesophageal reflux disease, inflammatory bowel disease, and ulcerative colitis.

To test for statistical significance between cohorts, Wilcoxon rank-sum tests were performed with corrections for multiple hypotheses where applicable when multiple comparisons are tested. Analyses were performed in R version 4.2.3.

## Public data set

Two sources of public data sets were used for the public data set analysis: the European Bioinformatics Institute MGnify portal (49) and the MicrobiomeHD data set collection, to form a cross-sectional survey data set (50). The data sets are described in more detail in Table 2.

The MGnify portal was searched using one or more of the following keywords— "diet," "GI indication," and "supplements." The resulting search results were further analyzed for relevance and shortlisted for data processing. Metadata were reviewed for relevance, namely that some of the samples in the study should be gut related and that metadata should contain information on participants who were healthy or had a gastrointestinal condition. Only gut-related samples in each study were downloaded. As taxonomic assignments are provided for both small subunit and large subunit, it was not clear which should be used, and therefore, the summary with the highest Operational Taxonomic Unit (OTU) count of the two was used. Analyses were performed on this data set using Python. To fit the format requirements of the *predict.py* tool, the following modifications were made: (i) removing "s" at the beginning of each OTU, (ii) removing all "Root;" occurrences, (iii) renaming "SampleID" to "OTU ID," and (iv) replacing ";" with "|" everywhere in the file. After running *predict.py* to derive MAPI and redox scores, the number of files in the output was verified with the number of samples in the taxonomic summary file.

The article by Duvallet et al. provided the microbiomeHD database (50), consisting of 29 case control studies containing 16S rRNA gene-based microbiota data sets. Studies can be downloaded individually as *.tar.gz files with top-level identifiers per data set. We used the metadata file and the 100% OTU tables with Latin taxonomic names assigned using the naive Bayesian classifier "RDP classifier" ($c = 0.5$). After minor changes were made, *predict.py* was run on the OTU tables for each data set.

### MAPI score calculation

For each microbiome sample, the MAPI score (8) was calculated as the natural logarithm of the ratio of the sum of the abundance of aerobic genera divided by the sum of the abundance of strict anaerobic genera. For Bifidobacteria, strict anaerobic taxa are defined at the species level. The list with strict anaerobic taxa was defined using the "List of Prokaryotes According to Their Aerotolerant or Obligate Anaerobic Metabolism" (8, 51).

### Keystone score calculation

The keystone score was calculated based on keystone species presented by Tudela et al. (15). The six keystone species used in this analysis were *Akkermansia muciniphila*, *Bifidobacterium longum*, *Christensenella minuta*, *Faecalibacterium prausnitzii*, *Methanobrevibacter smithii*, and *Ruminococcus bromii*. *Z*-scores were calculated from log-transformed relative abundances across each species. The keystone score is calculated by summing the *Z*-scores of six keystone species for each sample.

## ACKNOWLEDGMENTS

We would like to thank Ghislain Schyns for his helpful comments during manuscript preparation.

The financial support of dsm-firmenich/DSM Nutritional Products is acknowledged.

Contributor roles to the manuscript: Conceptualization T.N.M., S.J., A.S., J.W.S, Data curation J.P. , Formal analysis J.P., J.F.N, R.S., A.A., E.V.L.T., Project administration T.N.M., E.V.L.T., Visualization J.P., R.S., A.A., E.V.L.T., Writing – original draft J.K.B, Writing – review & editing J.K.B., J.P., A.S., T.N.M., J.W., E.V.L.T.

J.P. and S.J. are employees of Sun Genomics. R.S. and A.A. are employees of Solvuu. J.K.B. is a consultant for dsm-firmenich. A.S., J.W., J.F.N., E.V.L.T., and T.N.M. are employees of dsm-firmenich

## AUTHOR AFFILIATIONS

[1]Sun Genomics, Inc., San Diego, California, USA
[2]dsm-firmenich, Delft, The Netherlands
[3]Solvuu, Inc., New York, New York, USA
[4]Bird Scientific Writing, Wassenaar, The Netherlands
[5]dsm-firmenich, Kaiseraugst, Switzerland

## PRESENT ADDRESS

Jurgen F. Nijkamp, Heineken B.V., Zoeterwoude, the Netherlands
Tim N. Mak, Trilliome GmbH, Zürich, Switzerland

## AUTHOR ORCIDs

Joann Phan ⓘ https://orcid.org/0000-0001-5798-4416
Jonas Wittwer Schegg ⓘ http://orcid.org/0000-0002-8590-2657
Emiel Ver Loren van Themaat ⓘ http://orcid.org/0009-0005-1522-5856
Tim N. Mak ⓘ http://orcid.org/0009-0001-6161-2567

## AUTHOR CONTRIBUTIONS

Jonas Wittwer Schegg, Conceptualization, Funding acquisition, Project administration, Resources, Supervision.

## DATA AVAILABILITY

A full implementation of the MAPI score calculation is available via Bitbucket, https://bitbucket.org/dfsbioitteam/mapi_deploy/. The public data set can be accessed using the information provided in Table 2. Data from the Sun Genomics data set cannot be made open access due to subject confidentiality; however, de-identified data may be made available after reasonable request to the authors.

## ADDITIONAL FILES

The following material is available online.

### Open Peer Review

**PEER REVIEW HISTORY (review-history.pdf).** An accounting of the reviewer comments and feedback.

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
