## [Reviewer comments · Microbiology Spectrum]

Microbiology Spectrum

Gut health predictive indices linking gut microbiota dysbiosis with healthy state, mild gut discomfort and inflammatory bowel disease phenotypes using gut microbiome profiling

Tim Mak, Joann Phan, Suneer Jain, Rajkumar Sasidharan, Ashish Agarwal, Julia Bird, Anneleen Spooren, Jonas Wittwer Schegg, Emiel Ver Loren van Themaat, and Jurgen Nijkamp

Corresponding Author(s): Jonas Wittwer Schegg, DSM-Firmenich AG

Review Timeline:

Submission Date:	March 5, 2025
Editorial Decision:	April 29, 2025
Revision Received:	May 26, 2025
Accepted:	May 29, 2025

Editor: Se-Ran Jun

Reviewer(s): Disclosure of reviewer identity is with reference to reviewer comments included in decision letter(s). The following individuals involved in review of your submission have agreed to reveal their identity: Asaduzzaman Asad (Reviewer #2)

Transaction Report:

DOI: <https://doi.org/10.1128/spectrum.00271-25>

Re: Spectrum00271-25 (Gut health predictive indices linking gut microbiota dysbiosis with healthy state, mild gut discomfort and inflammatory bowel disease phenotypes using gut microbiome profiling)

Dear Dr. Jonas Wittwer Schegg:

Thank you for the privilege of reviewing your work. Below you will find my comments, instructions from the Spectrum editorial office, and the reviewer comments.

One of reviewers who happened to review your manuscript which was submitted to mSphere kept concerns regarding the study rationale with external cohorts and biological age that should be fully addressed, along with comments made by another reviewer.

Revision Guidelines

Sincerely,
Se-Ran Jun
Editor
Microbiology Spectrum

Reviewer #1 (Public repository details (Required)):

raw data a publicly available or available at request to corresponding author

Reviewer #1 (Comments for the Author):

This is my second revision of this ms submitted to mSphere. While study premise remains valuable, the authors pt by pt responses to my queries on conceptual gaps seem superficially argued. My recommendations aimed to enhance the work originality and depth in this competitive field. For example, dismissing my concerns regarding the study rationale with external cohorts and the lack of biological age data ("no information on biological age") overlooks an opportunity to discuss methodological limitations or propose frameworks for future research. Chronological age as a proxy remains an oversimplification, warranting deeper reflection-this was a missed opportunity to strengthen the discussion. Similarly, the analysis reiterates a well-established IBD-pathobiont associations, and I would have appreciated a deeper exploration of underdeveloped angles, such as disease-subtype stratification or longitudinal disease activity correlations with the mAPI/keystone score. While I initially encouraged the authors to discuss these gaps, I do not wish to unduly delay this work, as I believe it makes some contribution to the field.

Reviewer #2 (Comments for the Author):

The manuscript written by Phan et al., entitled "Gut health predictive indices linking gut microbiota dysbiosis with healthy state, mild gut discomfort and inflammatory bowel disease phenotypes using gut microbiome profiling," is a comparative methodological study where the authors compared MAPI and Keystones scales linking gut dysbiosis to gut discomfort and gastrointestinal diseases. Given the increasing interest in gut microbiota diagnostics, the topic is relevant. In the field of gut health prediction and microbiome research, this paper could be intriguing and useful. However, lack of specificity and well-defined comparative groups, the outcome seems clumsy and sometimes inconclusive.

Major comments:

1. Complete dependence on self-reported gastrointestinal conditions is a major limitation. This should be better addressed.
2. The use of conditions like mild discomfort, diseased, gastrointestinal diseases, IBD, healthy, etc., is not well defined and specific, which has made the outcome inconclusive.
3. Targeted metagenomics (e.g., 16S rRNA-based metagenomics) usually fails to define at the species level; then, how shotgun and targeted metagenomics results were combined is not clear.
4. The results from keystone scale are largely missing; should be proportionally mentioned with MAPI and comparatively illustrated.
5. Conclusions should be more concise and decisive; which scale is appropriate and where?

Other comments:

1. In title it is mentioned "healthy state, mild gut discomfort and inflammatory bowel disease". However, it is not described in the intro.
2. IBD vs Healthy was compared. What about other gastrointestinal diseases? Did they showed graded response too?
3. Line 287-296: Did the authors sequence the data? If yes, then why should they do that when they are using preexisting databases? If no, then this section should be omitted. Is it shotgun sequencing or targeted metagenomics? This section needs to be more specific and details.
4. What is the difference between Fig 5A and 5B?
5. Fig 1D can be presented as a table
6. Abbreviations are not properly used. Abbreviation should be mentioned with the elaboration on the first place, then it should be used uniformly.
7. Fig 4B needs X-axis title.

The manuscript written by Phan et al., entitled "**Gut health predictive indices linking gut microbiota dysbiosis with healthy state, mild gut discomfort and inflammatory bowel disease phenotypes using gut microbiome profiling**," is a comparative methodological study where the authors compared MAPI and Keystones scales linking gut dysbiosis to gut discomfort and gastrointestinal diseases. Given the increasing interest in gut microbiota diagnostics, the topic is relevant. In the field of gut health prediction and microbiome research, this paper could be intriguing and useful. However, lack of specificity and well-defined comparative groups, the outcome seems clumsy and sometimes inconclusive.

Major comments:

1. Complete dependence on **self-reported gastrointestinal conditions** is a major limitation. This should be better addressed.
2. The use of conditions like mild discomfort, diseased, gastrointestinal diseases, IBD, healthy, etc., is not well defined and specific, which has made the outcome inconclusive.
3. Targeted metagenomics (e.g., 16S rRNA-based metagenomics) usually fails to define at the species level; then, how shotgun and targeted metagenomics results were combined is not clear.
4. The results from keystone scale are largely missing; should be proportionally mentioned with MAPI and comparatively illustrated.
5. Conclusions should be more concise and decisive; which scale is appropriate and where?

Other comments:

1. In title it is mentioned "healthy state, mild gut discomfort and inflammatory bowel disease". However, it is not described in the intro.
2. IBD vs Healthy was compared. What about other gastrointestinal diseases? Did they showed graded response too?
3. Line 287-296: Did the authors sequence the data? If yes, then why should they do that when they are using preexisting databases? If no, then this section should be omitted. Is it shotgun sequencing or targeted metagenomics? This section needs to be more specific and details.
4. What is the difference between Fig 5A and 5B?
5. Fig 1D can be presented as a table
6. Abbreviations are not properly used. Abbreviation should be mentioned with the elaboration on the first place, then it should be used uniformly.
7. Fig 4B needs X-axis title.

Reviewer #1 (Public repository details (Required)):

Comment: *raw data a publicly available or available at request to corresponding author*

Response: The public dataset is publicly available. The privacy agreement for data from Sun Genomics customers did not include sharing data publicly and therefore we have to apply an exception to this dataset, in which it is only available upon reasonable request. Please see our revised data access statement.

Reviewer #1 (Comments for the Author):

Comment: *This is my second revision of this ms submitted to mSphere. While study premise remains valuable, the authors pt by pt responses to my queries on conceptual gaps seem superficially argued. My recommendations aimed to enhance the work originality and depth in this competitive field. For example, dismissing my concerns regarding the study rationale with external cohorts and the lack of biological age data ("no information on biological age") overlooks an opportunity to discuss methodological limitations or propose frameworks for future research. Chronological age as a proxy remains an oversimplification, warranting deeper reflection-this was a missed opportunity to strengthen the discussion. Similarly, the analysis reiterates a well-established IBD-pathobiont associations, and I would have appreciated a deeper exploration of underdeveloped angles, such as disease-subtype stratification or longitudinal disease activity correlations with the mAPI/keystone score. While I initially encouraged the authors to discuss these gaps, I do not wish to unduly delay this work, as I believe it makes some contribution to the field.*

Response: While we did have data concerning chronological age, biological age is a more difficult metric to calculate based on the data we collected from the microbiome of individuals. There is a model where biological age was derived from the metatranscriptomes of the gut microbiome in "An accurate aging clock developed from large-scale gut microbiome and human gene expression data" by Gopu et al. 2024. With the data type we collected in our study, we are unable to replicate this model.

In addition, biological age can be determined by analyzing various factors related to the age of an individual's cells and biological markers such as blood biomarkers, DNA methylation, gene expression, and others. The sample type collected in this study were metagenomic sequencing stool samples, which limits the predictive capability of calculating biological age in either of these models.

While we agree that biological age would add significant contributions to this work, we lack the additional biological data that would enable us to calculate biological age. We have included a small discussion of this important point in the manuscript.

Reviewer #2 (Comments for the Author):

The manuscript written by Phan et al., entitled "Gut health predictive indices linking gut microbiota dysbiosis with healthy state, mild gut discomfort and inflammatory bowel disease phenotypes using gut microbiome profiling," is a comparative methodological study where the authors compared MAPI and Keystones scales linking gut dysbiosis to gut discomfort and gastrointestinal diseases. Given the increasing interest in gut microbiota diagnostics, the topic is relevant. In the field of gut health prediction and microbiome research, this paper could be intriguing and useful. However, lack of specificity and well-defined comparative groups, the outcome seems clumsy and sometimes inconclusive.

Major comments:

Comment 1: Complete dependence on self-reported gastrointestinal conditions is a major limitation. This should be better addressed.

Response: We do agree that this is a major limitation to the study, and it is touched upon in the discussion, but we modified the text to expand on this limitation by stating that self-reported gastrointestinal conditions can be subjective (line 243-244).

Comment 2: *The use of conditions like mild discomfort, diseased, gastrointestinal diseases, IBD, healthy, etc., is not well defined and specific, which has made the outcome inconclusive.*

Response: While there is a limitation with self-reporting, there is a distinction between the mild and disease categories. The mild category included bloating, constipation, gassiness, and IBS and the disease category included diseases Crohn's disease, Celiac disease, GERD, IBD, and ulcerative colitis. The level of severity is generally different between symptoms like bloating, gassiness, and constipation compared to serious diseases like Crohn's disease and IBD. We agree that there could be varying degrees in severity in even symptoms like bloating, gassiness, constipation, and IBS, but an additional limitation is not identifying the levels of severity in those symptoms in this cohort.

Comment 3: *Targeted metagenomics (e.g., 16S rRNA-based metagenomics) usually fails to define at the species level; then, how shotgun and targeted metagenomics results were combined is not clear.*

Response: We agree that targeted metagenomics usually fails to define species level classification. That is the reason why we included a shotgun metagenomic dataset from Sun Genomics to compare healthy, mild, and diseased state individuals. The targeted dataset from public sources was not directly combined with the shotgun metagenomic results. Each dataset was kept separate in the analyses, but we reached the same conclusions with both datasets when assessing the health, mild, diseased, and health and IBD MAPI scores.

Comment 4: *The results from keystone scale are largely missing; should be proportionally mentioned with MAPI and comparatively illustrated.*

Response: We thank the reviewer for this valuable comment. We agree that including keystone score analyses alongside MAPI for the full public dataset would have strengthened the comparative aspect of the study. Unfortunately, we were unable to perform keystone scoring on the full public dataset because it is based on 16S rRNA sequencing data, which limits taxonomic resolution to the genus level. As the reviewer rightly noted, species-level resolution is essential for meaningful keystone analysis, and this was not feasible with the available data.

Comment 5: *Conclusions should be more concise and decisive; which scale is appropriate and where?*

Response: Given the scope of this study, it is difficult to conclude the use of these scales in a universal manner. While we do see significant differences in healthy, mild, and disease states, there is also overlap in these scores between states. There is a range in the scores in each state that make it difficult to confidently assign health, mild, or disease when assessing at the individual level. It is appropriate to assess at the population level because on average, people with varying levels of gastrointestinal health or distress are different from each other.

Other comments:

Comment 6: *In title it is mentioned "healthy state, mild gut discomfort and inflammatory bowel disease". However, it is not described in the intro.*

Response: Some background information has been added now to the introduction.

Comment 7: *IBD vs Healthy was compared. What about other gastrointestinal diseases? Did they showed graded response too?*

Response: While we did not compare Healthy to other distinct gastrointestinal diseases, we did compare Healthy to Diseased, which included a grouped population of gastrointestinal diseases. We found that there was a significant difference in the MAPI score between the Healthy, Mild, and Diseased cohorts.

Comment 8: *Line 287-296: Did the authors sequence the data? If yes, then why should they do that when they are using preexisting databases? If no, then this section should be omitted. Is it shotgun sequencing or targeted metagenomics? This section needs to be more specific and details.*

Response: Yes, we did sequence the Sun Genomics database. This was whole genome shotgun sequencing. We have clarified that in the text.

Comment 9: *What is the difference between Fig 5A and 5B?*

Response: The difference between Fig 5A and 5B are the comparisons between healthy subjects and IBD subjects in the Sun Genomics dataset in A and public datasets in B. The data indicate that in both datasets, the MAPI score for IBD is significantly higher than controls. This supports the case that the same result was found in both public and Sun Genomics databases.

Comment 10: *Fig 1D can be presented as a table*

Response: Figure 1D is presented as a table in Figure 1. Because it is related to the data in Figure 1B, we thought it would be a simpler visualization to include in the same figure rather than a separate table.

Comment 11: Abbreviations are not properly used. Abbreviation should be mentioned with the elaboration on the first place, then it should be used uniformly.

Response: We have scanned through the text and have made sure to use abbreviations correctly and uniformly.

Comment 12: Fig 4B needs X-axis title.

Response: We added an X-axis title to Fig 4B.

Re: Spectrum00271-25R1 (Gut health predictive indices linking gut microbiota dysbiosis with healthy state, mild gut discomfort and inflammatory bowel disease phenotypes using gut microbiome profiling)

Dear Dr. Jonas Wittwer Schegg:

Your manuscript has been accepted, and I am forwarding it to the ASM production staff for publication. Your paper will first be checked to make sure all elements meet the technical requirements. ASM staff will contact you if anything needs to be revised before copyediting and production can begin. Otherwise, you will be notified when your proofs are ready to be viewed.

Sincerely,
Se-Ran Jun
Editor
Microbiology Spectrum